# The Omni Scale Is Not Suitable for Assessing Self-Perceived Exertion during Physical Activity in Preschoolers

**DOI:** 10.3390/children10020238

**Published:** 2023-01-29

**Authors:** Carlos Ayán, José C. Diz, Silvia Varela, Miguel A. Sanchez-Lastra

**Affiliations:** 1Department of Special Didactics, University of Vigo, 36005 Pontevedra, Spain; 2Well-Move Research Group, Galicia Sur Health Research Institute (IIS Galicia Sur), Sergas-UVIGO, 36312 Vigo, Spain

**Keywords:** exercise intensity, assessment, children, self-perception, field-based tests

## Abstract

Introduction: We aimed to identify the utility, reliability, and validity of an adapted version of the OMNI self-perceived exertion (PE) rating scale in preschoolers. Population and methods: Firstly, 50 (mean age ± standard deviation [SD] = 5.3 ± 0.5 years, 40% girls) performed a cardiorespiratory fitness (CRF) test twice, with a one-week interval between assessments, and rated their PE either individually or in groups. Secondly, 69 children (mean age ± SD = 4.5 ± 0.5 years, 49% girls) performed two CRF tests, separated by a one-week interval, twice and rated their self-PE. Thirdly, the heart rate (HR) of 147 children (mean age ± SD = 5.0 ± 0.6 years, 47% girls) were compared against self-rated PE after finishing the CRF test. Results: Self-assessed PE differed when the scale was administered individually (e.g., 82% rated PE with 10) or in groups (42% rated PE with 10). The scale showed poor test–retest reliability (ICC:0.314-0.031). No significant associations were found between the HR and PE ratings. Conclusions: An adapted version of the OMNI scale was found not suitable for assessing self-PE in preschoolers.

## 1. Introduction

In preschoolers, cardiorespiratory fitness is associated with cardiometabolic health, premature cardiovascular disease, academic achievement, and mental health [1]. However, over the past decades, cardiorespiratory fitness levels have declined, and this is largely explained by an increase in sedentary time and decreased levels of moderate-to-vigorous physical activity [2].

This fact emphasizes the importance of promoting and monitoring the participation in physical activity of at least moderate intensity in this population. In order to monitor physical activity intensity, the rate of perceived exertion, a psychophysical marker of the intensity of the exercise response, seems to be a useful resource [3].

Perceived exertion (PE) refers to the level of effort, discomfort, and fatigue experienced during physical exercise [4]. The rate of PE, which is usually assessed using self-assessment scales, allows the intensity of the exercise to be quantified, monitored, and regulated [5]. These scales were originally designed for adult populations and were not considered suitable for young children; therefore, specific self-assessment scales were developed to suit children’s cognitive abilities and to improve the accuracy of PE measurements [6]. Although there are different scales for assessing PE in children that show accurate psychometric properties (e.g., OMNI, EPINFANT, CRET) [7,8], little research has been conducted on the reliability and validity of these scales when administered in preschoolers.

The application of self-PE scales in the paediatric setting is rather limited [9], so it is of interest to determine whether PE self-assessment scales are suitable for pre-school children. The administration of these scales would allow not only the monitoring of exercise intensity during structured training programmes, but also the assessment of fitness levels and the identification of the onset of fatigue during physical exertion [10].

Considering this, the aim of this investigation was to identify the utility, test–retest reliability, and convergent validity of the OMNI self-PE rating scale through a series of studies with preschoolers.

## 2. Materials and Methods

### 2.1. Participants

Subjects were selected from healthy Spanish children recruited from three kindergartens in rural areas of northern Spain. Inclusion criteria for participation were: (a) to be enrolled in the first or second level of the first stage of the Spanish education system and (b) not to present diseases that could impede the performance of the field tests proposed in the study. Written informed consent was obtained from the school principals and parents or guardians of the children enrolled. The local ethics committee reviewed the protocols in advance and raised no objections.

### 2.2. Measurements

#### 2.2.1. Self-Perceived Exertion

Self-PE rating was assessed using an adapted version of the OMNI scale for children [11]. This scale has a developmentally indexed category format that contains four pictorial and six verbal descriptors positioned along a comparatively narrow numerical response range from 0 to 10, with 0 representing ‘‘not tired at all’’ and 10 representing ‘‘very, very tired”. For this research, the walking/running version of the OMNI scale was selected, which is a valid tool for assessing self-PE in children aged 6 to 13 years [12]. To create meaningful expressions of exertion for the preschoolers, a kindergarten teacher developed four pictorial descriptors illustrating a young child experiencing varying levels of exertion while climbing a hill.

#### 2.2.2. Exercise Test

Two cardiorespiratory fitness field tests, the Mini-Cooper (MC) and the three-minute shuttle run (3MSR), were used as self-paced physically demanding tasks to induce physiological stress and fatigue. Both tests have shown adequate reliability values (intraclass correlation coefficient [ICC] of 0.81 to 0.94) when performed by pre-school children [13]. For the MC, children were asked to move around a rectangle mark on the floor (with dimensions of 9 × 18 m) as fast as possible for a period of 6 min. For the 3MSR, two poles (1.5 m in height) were placed 10 m apart to create a straight 10 m long running track. Children were asked to run from one side to the other, turn around the pole, and then return to the starting point. Walking was allowed if needed in both tests. The score was the distance (m) covered in 6 and 3 min, respectively.

#### 2.2.3. Anthropometry

Weight was measured to the nearest 0.1 kg using a Tefal digital scale (type PP1200VO) with the child dressed in light clothing and without shoes. Height was measured to the nearest millimeter using a field stadiometer (Seca 220). The body mass index (BMI) was calculated as the body weight (kg) divided by the height squared (m^2^), and participants were classified according to age-specific percentiles into underweight (below the 3rd percentile), normal weight (3rd to 85th percentiles), overweight (85th to 97th percentiles), and obese (beyond the 97th percentile) [14].

All measurements were performed by the kindergarten teacher and two final year students of the Degree in Early Childhood Education, who were familiar with the testing protocols. All tests were conducted in groups of 10 children during the two weekly sessions devoted to psychomotricity, a core subject of the pre-school academic curriculum in Spain.

### 2.3. Procedures

For the purpose of this research, three different procedures were developed, in which the exercise test and the self-PE scale were administered to three different samples of participants. Each procedure started with a familiarization session one week before the start of the test [15]. During this session, the OMNI scale was described and explained following established guidelines [11], and children were also introduced to the correct performance and execution of the exercise tests. Anthropometric measurements were also performed.

#### 2.3.1. Study 1–Simultaneous Administration

To identify whether children’s self-PE levels were influenced by that of their classmates, 50 preschoolers (mean age: 5.30 ± 0.46 years; BMI: 17.60 ± 1.38 kg/m^2^; 40% girls) performed the 3MSR test and rated their self-PE on the scale twice, with an interval of one week between the two assessments. During the first session, the children performed the test individually and were asked about how they felt according to the self-PE rating scale immediately after completion. During the second session, the 3MSR was administered through groups of five children, who indicated their self-PE levels together after the test was completed.

#### 2.3.2. Study 2–Reliability

To analyse the test-retest reliability of the scale, 69 preschoolers (mean age: 4.51 ± 0.50 years; BMI: 16.18 ± 1.38 kg/m^2^; 49.3% girls) rated their level of PE after completing both exercise tests, twice, through a period of three weeks. During the first session of the first week, the children performed the MC. The 3MSR was completed during the second session. Children were asked to provide a PE rating just after completing each test. After a one-week interval, the procedure was repeated [15]. Children were not reminded of their previous ratings to avoid bias.

#### 2.3.3. Study 3–Validity

To analyse the concurrent validity of the scale, 147 children (mean age: 5.04 ± 0.60 years; BMI: 16.31 ± 1.84 kg/m^2^; 46.9% girls) performed the 3MSR while wearing a heart rate (HR) monitor (Polar RS400, Kempele, Finland), and HR was registered immediately after the end of the test with the HR monitor connected via Bluetooth to an iPad Air 2. The association between the self-PE level immediately after the exercise tests and the HR was used as a test of the concurrent validity of the scale [8]. Based on the preliminary results observed in Study 1 and Study 2, the scale underwent two modifications, aiming at a more accurate self-PE rating. First, an alteration in the colours of the pictures was used to relate to different levels of exercise intensity [16]. Secondly, we reviewed previous findings indicating that preschoolers can discriminate between three different levels of exertion when using self-PE scales [17]. Considering this and that most of our children tended to discriminate only the extreme values, the four pictorial and six verbal descriptors of the scale were positioned along a numerical response range of 1 to 5.

### 2.4. Statistical Analysis

Data are expressed as mean ± SD for quantitative variables, or as n (%) for qualitative variables. We assessed the normality of the data using the Kolmogorov–Smirnov test and compared continuous variables with independent data Student’s t-test or ANOVA, or with Mann–Whitney U or Kruskal–Wallis test if they did not have normal distribution. Test–retest reliability was assessed by calculating the ICC with 95% confidence intervals. Values above 0.9 were considered excellent, between 0.75 and 0.9 good, between 0.5 and 0.75 moderate, and below 0.5 poor [18]. To analyse the validity of the self-PE rating scales, we calculated Spearman’s rank correlation coefficient between the scales and the HR or distance reached in the tests. In addition, as sensitivity analyses, due to previous evidence suggesting that body composition may influence children’s perception of fatigue, we repeated the analyses stratified by BMI [19,20]. All analyses were performed using SPSS Statistical Software and a two-sided P value of less than 0.05 indicated statistical significance.

## 3. Results

A total of 50, 69, and 147 children of four to six years old were included in studies one, two and three, respectively. Descriptive characteristics of the included participants are shown in Table 1.

### 3.1. Study 1: Simultaneous Administration

Important differences were observed comparing how children rated their level of PE when asked individually versus in a group. In the first case, 82% of the participants selected “10” as a numerical response in the adapted OMNI scale, while 8% selected “0”. Diverse numerical responses were selected by the remaining 10%. In the second case, 42% and 16% of the children selected “10” or “0”, respectively, as their numerical response. The influence of the group on self-PE rating can be observed in Figure 1. In half of the groups, all five children provided the same answer, while in three groups, four out of the five children rated their self-PE identically. The mean total distance covered in the 3MSR was 278.40 ± 57.41 m (first assessment) and 277.60 ± 65.86 m (second assessment). No statistically significant associations were observed between the 3MSR and the adapted OMNI scale when children rated their self-PE either individually (in the first assessment) or in groups (in the second assessment).

### 3.2. Study 2: Reliability

Figure 2 depicts the distribution of the numerical responses selected by the children in the OMNI scale during the first (test) and second (retest) administration of both the 3MSR and the MC tests. Globally, a dichotomous pattern was observed by which children were either “very, very tired” (41.6% of them selected “10” as numerical response) or “not tired” (28.2% of them selected either “0” or “1” as a numerical response). When comparing the numerical ratings given by the children after finishing the 3MSR, the adapted OMNI scale showed a poor test–retest reliability (ICC:0.314; 95% CI: 0.086–0.511). In the case of the MC reliability was even lower (ICC: 0.031; 95% CI: 0.199–0.026).

Figure 3 shows in three-dimensional form the number of children answering each item on the OMNI scale in the test and retest for both cardiorespiratory fitness tests. The larger number of children in the “four corners” that can be seen in the figure shows the pattern of answering extreme values.

No significant differences were observed when comparing the distance covered in the fitness tests, neither in the test (3MSR, 297.97 ± 44.84 m; MC, 610.03 ± 112.47 m) nor the retest (3MSR, 304.49 ± 43.23 m; MC, 598.30 ± 128.68 m). Again, no significant associations were observed between the distance covered in the field-based tests and the children’s rating of self-PE. No effect of BMI on the self-perceived fatigue was observed.

### 3.3. Study 3: Validity

The numerical responses given by the children after finishing the 3MSR are shown in Table 2. Again, extreme values were reported, although a greater diversity of percentages was observed in comparison with the previous two studies. No effect of BMI on the self-perceived fatigue was observed.

Table 3 shows mean and delta HR values. Despite the observed increase in HR, no significant associations were found between this physiological variable and the self-PE ratings.

## 4. Discussion

In this study we aimed to provide information on the usefulness and psychometric properties of the OMNI self-PE rating scale in preschoolers. The results obtained question the usefulness of these measurement instruments when administered in young populations.

One of the reported advantages of using scales for assessing PE is that these allow for a simultaneous administration (i.e., a trained observer may be able to rate the EF of several individuals in a short period of time) [21]. The results obtained in our first study are contrary to this idea. In fact, important differences were observed when comparing how children rated their self-PE when asked individually or in a group. These findings could be partially affected by “social conformity” (i.e., the act of giving up one’s own beliefs by adopting the behaviour of the majority) [22]. Indeed, very young children have been observed to spontaneously change their private opinions under the implicit social influence of their peers [23]. Thus, it seems that self-PE rating scales should be administered individually to obtain solid results.

In the context of aerobic exercise, the utility of PE ratings in children and adolescents remains unclear due to a paucity of evidence and inconsistencies across studies [9 Kasai 2021-10-]. To expand the existing scientific knowledge on this topic, we used two aerobic field tests to determine the degree of reliability and validity shown by a self-PE rating scale when administered to preschoolers. The results of studies two and three suggest that an adapted version of the OMNI scale was not suitable for assessing self-PE in children under six years of age.

We observed that preschoolers tended to report extreme values when asked to rate their self-PE levels. This result is consistent with the idea that, at this early stage of development, effort is perceived as “easy” or “hard” according to their physiological maturity [24]. However, in our second study, when comparing the numerical scores of the test and retest, it was observed that only one third of the children provided the same answer when asked about their self-PE, indicating poor reliability. Self-PE rating scales have shown good reliability values in children aged 8–12 years [9]. We only found one study that informed about the reliability of these scales when performed by preschoolers, with coefficients ranging from very poor (ICC = 0.17) to good (ICC = 0.77) [16]. These differences could be explained by the fact that the authors used an intermittent and progressive exercise test and included six year olds as part of their sample.

To identify convergent validity, we assumed a relatively stable increase in perceived exertion during high-intensity, self-paced activity [25]. Indeed, in our third study, when comparing baseline and final HR values, a considerable increase in exertion level was confirmed in the sample [26]. Despite this, no significant correlations were observed between HR and self-PE. Moreover, no associations were found between the distance covered in the field-based tests and the level of self-PE in any of the three studies. These findings imply that the relationship between intensity indicators (i.e., fitness test score) and physiological variables (i.e., HR) that support the application of the rating of PE in sport and exercise in previous studies [9], was almost non-existent in our sample of preschoolers.

Self-PE scales have shown accurate levels of validity in children aged eight years and older [8], while it has been argued that, after the age of five years, cardiorespiratory factors are progressively involved and allow six-year-old children to estimate PE accurately [17]. Despite this, it has been suggested that caution should be exercised when interpreting self-PE scale reports from children under eight years of age [27]. Our results reinforce this assumption.

Little research has assessed the validity of PE self-assessment scales in preschoolers using field aerobic tests as a self-paced task. Williams et al. [24] found significant associations (r = 0.73) between self-PE ratings and the HR in a group of 28 preschoolers performing an incremental exercise protocol. Similar results (r = 0.78) were obtained by Groslambert et al. [16], although the sample (n = 13) included children aged five and six years. Differences between the findings of both studies and the present research can be partially explained by the following aspects. Firstly, we used a continuous exercise test and the children were asked to rate their PE just after finishing it. However, in the two studies mentioned above, an incremental test was performed in which children were asked to stop and rate their PE at the end of each step. In addition, both studies included very few participants, while we tested a total of 266 preschoolers, which was a considerably larger sample compared to previous studies [8,9,17]. Finally, it should be noted that the accuracy of the OMNI scale for self-PE rating in children might not be the best due to its verbal descriptors. In this scale, a baseline level of “tiredness” is assumed, which can be interpreted as a sign of fatigue, rather than indicator of exertion [9]. The findings of the aforementioned investigations support this assumption, as Williams et al. [24] used a scale based mainly on simple verbal descriptions such as “easy” or “hard” (CERT scale) while Groslambert et al. [16] administered a scale that only included pictures (RPE-C scale).

There is a paucity of studies that have focused on the relationship between body composition and fatigue in children [19]. Our data expands the existing body of knowledge by the inclusion of information about the influence of BMI on self-PE. The association between body composition and fatigue has been previously analysed in the literature. Self-reported fatigue has been found to be associated with higher BMI among adults [28]. Similarly, it has been reported that children with obesity experienced significantly higher levels of fatigue compared to healthy children [29]. In the present investigation, BMI did not influence self-P SP-E among preschoolers. This result echoes the findings by Laurent et al. (2019) [19], who compared ratings of PE in a group of children who performed submaximal cycling sessions. The authors found that overweight children reported similar levels of differential exertion compared to non-overweight children while cycling at 70% of HRpeak. In contrast, Marinov et al. (2002) [30] reported that obese children rated PE significantly higher than non-overweight children while performing an incremental test. Notably, Marinov et al. (2002) [30] used body fat percentage as a marker of obesity, which may be more adequate when determining whether body composition is predictive of self-PE among young populations [20].

The results of this study add valuable information to the current body of knowledge on the usefulness of self-PE rating scales in preschoolers. We have overcome some of the most frequent methodological limitations in this field of research, such as samples of small sizes and consistent testing mostly of boys, under-representing girls [9]. However, some weaknesses must also be acknowledged. First, we used an estimation paradigm approach in which children were asked about their PE after finishing a self-paced task. Thus, we did not analyse whether children were able to regulate their own exercise intensity using a PE scale. Secondly, we used field-based tests of fitness, rather than performing an exercise protocol under laboratory conditions, which is considered the gold standard method for identifying PE objectively [1]. Thirdly, we only conducted one test to assess the convergent validity of the self-PE rating scale, whereas performing two tests would provide more solid and robust results. Finally, we tested the validity and reliability of the scale through the performance of a physical test of vigorous intensity. Whether psychometric properties differ at lower exercise intensities (e.g., moderate or light) requires further investigation.

## 5. Conclusions

An adapted version of the OMNI scale for identifying self-perceived physical exertion among preschoolers exhibited poor test–retest reliability and convergent validity. Children seem to answer differently to this scale depending on whether they are asked to rate their physical effort individually or in a group.

## Figures and Tables

**Figure 1 children-10-00238-f001:**
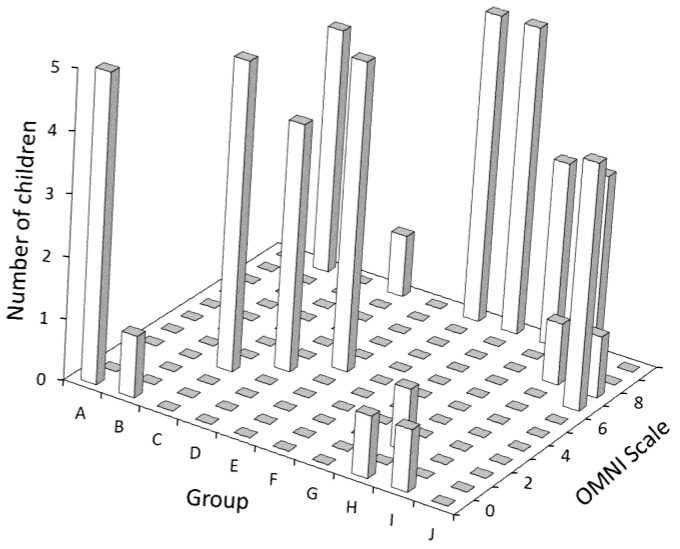
Results of the OMNI Scale administered in groups of five children, divided by group.

**Figure 2 children-10-00238-f002:**
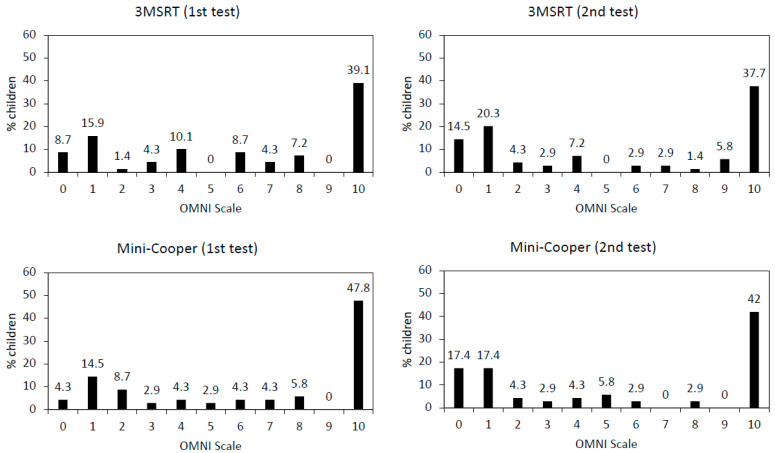
Results of the OMNI Scale for the three-minute shuttle run test (3MSRT) and the Mini-Cooper test, administered twice each.

**Figure 3 children-10-00238-f003:**
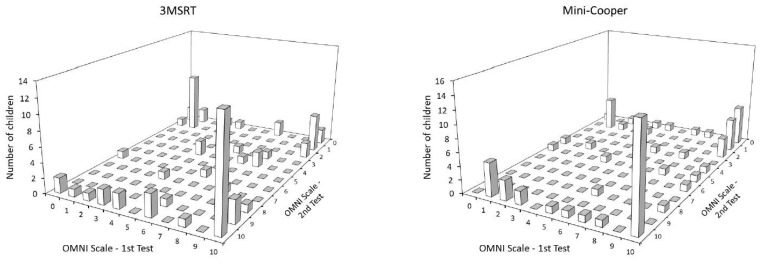
Comparison of the results of the test and retest of the 3MSRT and Mini-Cooper.

**Table 1 children-10-00238-t001:** Descriptive characteristics of the included sample.

		Total Sample (n = 266)	Study 1 (n = 50)	Study 2 (n = 69)	Study 3 (n = 147)
Sex					
	Girls	53.8	60.0	50.7	53.1
	Boys	46.2	50.0	49.3	46.9
Age (mean ± SD years)		4.9 ± 0.6	5.3 ± 0.5	4.5 ± 0.5	5.0 ± 0.6
Age strata					
	4 years old	26.3	0.0	49.3	24.5
	5 years old	51.1	70.0	50.7	44.9
	6 years old	22.7	30.0	0.0	30.6
Weight (mean ± SD kg)		20.8 ± 3.2	20.5 ± 2.3	19.7 ± 2.6	21.3 ± 3.62
Height (mean ± SD cm)		112.1 ± 6.5	108.3 ± 6.5	110.32 ± 5.0	114.1 ± 6.3
BMI (mean ± SD kg/m^2^)		16.5 ± 1.7	17.6 ± 1.4	16.2 ± 1.4	16.3 ± 1.8
BMI strata					
	Underweight	0.8	0.0	1.4	0.7
	Normal weight	60.2	32.0	66.7	66.7
	Overweight	28.2	46.0	27.5	22.4
	Obese	10.9	22.0	4.3	10.2

Values are percentage unless stated otherwise. BMI: body mass index; SD: standard deviation. Participants were classified according to their body mass index following the World Health Organization’s criteria using age-specific percentiles as follows: below the 3rd percentile for underweight, between 3rd to 85th percentiles for normal weight, between 85th and 97th for overweight, and beyond 97th percentile for obesity.

**Table 2 children-10-00238-t002:** Distance covered in the 3-minute shuttle run test and OMNI scale ratings across sex, age, and body mass index strata.

		3-Minute Shuttle Run Test (m)		OMNI Scale Rating (Range 1 to 5)		
		Mean (SD)	*p*	Mean (SD)	Median (IQR)	*p*
Sex	Boys	300.59 (28.87)	<0.001	2.21 (1.62)	1 (1; 3.25)	0.012
	Girls	275.44 (35.48)		2.81 (1.61)	2 (1; 4.5)	
Age (years)	4	283.75 (36)	0.449	2.47 (1.61)	1 (2; 4)	0.984
	5	288.34 (66)		2.52 (1.64)	1 (2; 4)	
	6	293.47 (45)		2.47 (1.70)	1 (2; 4.5)	
Body Mass Index	Underweight *	330	0.319	1	1	0.331
	Normal weight	290.06 (34.53)		2.55 (1.64)	2 (1; 4)	
	Overweight	289.41 (28.18)		2.64 (1.75)	2 (1; 5)	
	Obese	276.37 (44.45)		1.87 (1.30)	1 (1; 3)	

IQR: Interquartile range; SD: Standard deviation. Participants were classified according to their body mass index following the World Health Organization’s criteria using age-specific percentiles as follows: below the 3rd percentile for underweight, between 3rd to 85th percentiles for normal weight, between 85th and 97th for overweight, and beyond 97th percentile for obesity. * Only one participant was classified as underweight.

**Table 3 children-10-00238-t003:** Heart rate results across sex, age, and body mass index strata.

		Resting (bpm)		Final (bpm)		Delta (bpm)		Delta (%)	
		Mean (SD)	*p*	Mean (SD)	*p*	Mean (SD)	*p*	Mean (SD)	*p*
Sex	Boys	111.5 (11.01)	0.537	193.27 (7.5)	0.758	81.77 (12.54)	0.464	75.05 (19.24)	0.599
	Girls	110.43 (9.67)		193.68 (8.53)		83.25 (11.74)		76.61 (16.14)	
Age (years)	4	106.56 (10.41)	0.006	194.67 (8.33)	0.550	88.11 (11.5)	0.002	84.28 (19)	0.002
	5	111.53 (10.52)		193.29 (8.25)		81.76 (12.47)		74.78 (17.56)	
	6	113.78 (9.15)		192.76 (7.32)		78.98 (10.77)		70.46 (14.82)	
Body Mass Index	Underweight	115	0.459	194	0.258	79	0.828	68.7	0.629
	Normal weight	110.82 (9.51)		192.76 (7.34)		81.94 (11.85)		75.27 (17.3)	
	Overweight	109.73 (13.54)		193.88 (9.45)		84.15 (13.41)		78.95 (20.64)	
	Obese	114.73 (7.7)		197.13 (8.2)		82.4 (12.13)		72.63 (14.81)	

BPM: beats per minute; SD: Standard deviation. Participants were classified according to their body mass index following the World Health Organization’s criteria using age-specific percentiles as follows: below the 3rd percentile for underweight, between 3rd to 85th percentiles for normal weight, between 85th and 97th for overweight, and beyond 97th percentile for obesity.

## Data Availability

All data are available upon request to the corresponding author.

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
