# Peer review of "The Omni Scale Is Not Suitable for Assessing Self-Perceived Exertion during Physical Activity in Preschoolers"

_children, 2023, doi:10.3390/children10020238_

Round 1

Author Response

English language and style are fine/minor spell check required.

A: The language and style has been reviewed and modified accordingly.

1. The abstract is concise and straight forward. However, it would be informative to provide more detailed information in the Methods (age of participants, test-retest interval etc.) and Results.

A: Thank you for the suggestion. The abstract has been edited to expand on this information.

2. No descriptive characteristics of the subjects were shown in the manuscript. Please add one table to show this basic information.

A: We have included a new table (Table 1) with this information.

3. The validation study was conducted using the Three-minute Shuttle Run, which could be exercise of vigorous intensity as indicated by the HR. Thus, it is unknown about the reliability and validity of the scale for exercise of low or moderate intensity. I would suggest the authors to comment this point in the discussion (for example, when comparing to previous studies, and in the limitation).

A: Thank you for the suggestion. We agree that this might be an interesting aspect. However, we did not find any previous studies reporting information on children’s self-perceived exertion to physical activity of moderate intensities or lower. We have included this in the limitations.

Reviewer 2 Report

This manuscript has investigated utility, reliability, and validity of an adapted version of the OMNI self-perceived exertion (PE) rating scale in preschoolers. The authors found poor test-retest reliability and convergent validity of the Omni Scale among preschoolers. Findings also suggested the current scale is not valid for one given exercise (the Three-minute Shuttle Run).  In general, this study is well conducted. Despite this, the manuscript has several issues that need to be addressed.

1. The abstract is concise and straight forward. However, it would be informative to provide more detailed information in the Methods (age of participants, test-retest interval etc.) and Results.

2.  No descriptive characteristics of the subjects were shown in the manuscript. Please add one table to show this basic information.

3.   The validation study was conducted using the Three-minute Shuttle Run, which could be exercise of vigorous intensity as indicated by the HR. Thus, it is unknown about the reliability and validity of the scale for exercise of low or moderate intensity. I would suggest the authors to comment this point in the discussion (for example, when comparing to previous studies, and in the limitation).

Author Response

(The authors gave the same response as above.)

Reviewer 3 Report

English is somewhere difficult to understand. Need to be reviewed.

In ther results, results are presented according to children's body weight, e.g. overweight and  obese. It need to be described in the methods (participants) and in the aims. Also, how you chosed the cut off points for overweight and obese must be stated and references must be provided. 

Table 2 must be reviewed and ordered, there are some data that are out the columns.

Author Response

Reviewer 3:

English is somewhere difficult to understand. Need to be reviewed.

A: The language and style has been reviewed and modified accordingly.

In ther results, results are presented according to children's body weight, e.g. overweight and  obese. It need to be described in the methods (participants) and in the aims. Also, how you chosed the cut off points for overweight and obese must be stated and references must be provided. 

A: Thank you for the suggestions. We did not include the body mass index-stratified analysis in our aims, but as a sensitivity analysis instead, because there is some evidence to suggest that children's perception of fatigue may be influenced by body composition (e.g., PMIDs 30842711 and 29140314), but more precise indicators of body composition than BMI are probably required to appropriately assess this aspect (PMID: 29140314). We have modified the methods accordingly, to expand on this information. We have also included more information in Methods’ section and as footnotes in the tables to clearly describe how we classified participants based on their  body mass index. We have added a paragraph discussing the influence of BMI in self-perceived exertion.

Table 2 must be reviewed and ordered, there are some data that are out the columns.

A: Table 2 has been reviewed and all data should fit the columns appropriately now.

We have also highlighted in the table that there is only one participant classified as underweight, thus the lack of decimals and standard deviation in the respective cells and columns.